# Acute Acalculous Cholecystitis Associated with Abscesses—An Unknown Dual Pathology

**DOI:** 10.3390/biomedicines11020632

**Published:** 2023-02-20

**Authors:** Cristina Gluhovschi, Florica Gadalean, Silvia Velciov, Ligia Petrica, Ciprian Duta, Mircea Botoca, Daniela Cipu

**Affiliations:** 1Department of Internal Medicine II, Division of Nephrology, “Victor Babeș” University of Medicine and Pharmacy, County Emergency Hospital Timisoara, Eftimie Murgu Sq. No. 2, 300041 Timișoara, Romania; 2Centre for Molecular Research in Nephrology and Vascular Disease, Faculty of Medicine, “Victor Babeș” University of Medicine and Pharmacy, Eftimie Murgu Sq. No. 2, 300041 Timișoara, Romania; 3Department X Surgery II, Division of Surgery II, “Victor Babeș” University of Medicine and Pharmacy, County Emergency Hospital Timisoara, Eftimie Murgu Sq. No. 2, 300041 Timișoara, Romania; 4Department XV Orthopedics-Traumatology, Urology, Radiology and Medical Imaging, Division of Urology, “Victor Babeș” University of Medicine and Pharmacy, County Emergency Hospital Timisoara, Eftimie Murgu Sq. No. 2, 300041 Timișoara, Romania; 5Department XV Orthopedics-Traumatology, Urology, Radiology and Medical Imaging, Division of Radiology and Medical Imaging, “Victor Babeș” University of Medicine and Pharmacy, County Emergency Hospital Timisoara, Eftimie Murgu Sq. No. 2, 300041 Timișoara, Romania

**Keywords:** renal abscess, acute acalculous cholecystitis, diagnostic imaging

## Abstract

(1) Introduction and Aims: Little is known about the relationship between renal pathology and gallbladder pathology, although the two organs (the gallbladder and the right kidney) are in close proximity to one another. If a renal abscess disseminates, the gallbladder would be one of the secondary organs involved. As the bile provides a favorable environment for the development of pathogenic germs, it allows for the development of acute cholecystitis, even if calculi are absent, thus resulting in the development of acute acalculous cholecystitis. The aim of our study was to analyze the association between acute acalculous cholecystitis (AAC) and renal abscesses. (2) Methods: A department-wide retrospective cohort observational study including 67 patients with renal abscesses, with a mean age of 34.5+/−16.21 years and with five males and 62 females, was conducted. All of the patients were examined by an abdominal ultrasound. The lab tests included CBC with differential liver enzymes and serum bilirubin (in order to assess alterations in the liver function which can be associated with AAC) and serum creatinine (in order to assess the renal function). Blood culture and urine culture tests were also performed. (3) Results: Of the 67 patients with renal abscesses, eight (11.94%) were associated with acute cholecystitis: four cases (5.97%) of acalculous cholecystitis and four cases (5.97%) of calculous cholecystitis, two of which presented biliary sludge (acute micro-calculous cholecystitis). All four cases of acute acalculous cholecystitis presented with sepsis, and there was one case of septic shock at onset. We did not observe an impairment in renal function in the patients presenting with acute acalculous cholecystitis, and hepatic impairment was inconstant and moderate. All of the cases had a favorable outcome after a prompt initiation of intensive antibiotic therapy; both the renal abscess and the acute acalculous cholecystitis receded without further complications. (4) Conclusions: The association of acute acalculous cholecystitis with renal abscesses could be related to the possibility of germ dissemination from the infectious focus. In the case of a renal abscess, careful clinical, lab, and imaging exams of the gallbladder are recommended in order to ensure early therapeutic intervention in the event of an association with acute acalculous cholecystitis.

## 1. Introduction

The relationship between renal pathology and the pathology of the digestive system has often been emphasized. The relationship between renal pathology and the pathology of the gallbladder is less well-known, although both infections of the urinary tract and of the gallbladder are frequent. Moreover, the right kidney and the gallbladder are in close proximity, which could allow for an infection of the kidney to spread to the hepatobiliary system and vice versa.

The present paper aims to present the relationship between renal abscesses and acute cholecystitis, mainly acute acalculous cholecystitis (AAC), which is a form of acute cholecystitis which can have a severe outcome. This disease is most frequently encountered in critically ill patients (Huffman and Schenker) [1].

AAC has been evidenced after surgery, trauma, burns, shock, total parenteral nutrition, and prolonged fasting.

Despite the recent progress made in magnetic resonance imaging techniques, AAC still presents diagnosis difficulties in case of critically ill patients. AAC patients can experience severe outcomes accompanied by complications such as gangrene and perforations, which impose an appropriate surgical intervention (cholecystectomy).

Renal abscesses represent a severe disease of the kidney. They can be accompanied by the elimination (dissemination) of germs via the bloodstream. These germs can colonize the gallbladder, thus infecting the bile and producing secondary acute cholecystitis.

An infection at the gallbladder level can occur in the presence of biliary lithiasis (a favoring factor), producing acute calculous cholecystitis, or—in the absence of calculi—causing AAC.

Since renal abscesses can be accompanied by a severe clinical picture similar to the ones encountered in critically ill patients, the aim of our study was to analyze the association of renal abscesses with AAC.

This combined pathology is not mentioned as such in the literature, and the clinical presentation and therapeutic line of conduct for this association are important for medical practice.

## 2. Methods

A department-wide retrospective cohort observational study was conducted including 67 patients with renal abscesses, with a mean age of 34.5+/−16.21 years, and consisting of five males and 62 females who were admitted to our academic Nephrology Department over a time frame of 4 years. All patients were examined by an abdominal ultrasound. Lab tests included CBC with differential liver enzymes and serum bilirubin (in order to assess alterations in the liver function which can be associated with AAC) and serum creatinine (in order to assess renal function). Blood culture and urine culture tests were also performed.

ACC diagnosis was established on clinical grounds and was based on the ultrasound criteria. ACC diagnosis was based on the absence of biliary lithiasis and/or of sludge on the ultrasound.

Other criteria for diagnosing AAC were:-Right upper quadrant pain which intensified under pressure with the ultrasound probe;-Thickening of the gallbladder wall ≥ 3 mm;-Gallbladder fluid.

The presence of two of these criteria defined ACC diagnosis.

It should be mentioned that the altered clinical condition of the patients required prompt therapy with antibiotics after the appropriate blood cultures were withdrawn, this being the reason why the blood cultures were not repeated thereafter.

The ultrasound exam was performed on a repetitive basis in order to monitor the outcomes. Three patients underwent computerized tomography (CT).

All patients were administered an intravenous combination of two antibiotics intravenously. Thus, three cases of AAC were administered carbapenem with an aminoglycoside, and one case received a 3rd generation cephalosporin with an aminoglycoside.

The cases of acute cholecystitis which presented biliary sludge were also assessed. These are not included in the diagnosis of AAC. We have considered them as a particular group: acute micro-calculous cholecystitis.

The cases of acute cholecystitis presenting lithiasis (calculous cholecystitis) were also analyzed. 

## 3. Results

In total, eight (11.94%) of the 67 patients with renal abscesses were associated with acute cholecystitis: Four cases (5.97%) were associated with acalculous cholecystitis;Four cases (5.97%) were associated with calculous cholecystitis, of which two presented biliary sludge (micro-calculous cholecystitis).

All four cases of acute acalculous cholecystitis presented sepsis, and one case presented septic shock at onset. We did not observe impairment in renal function in the patients presenting with acute acalculous cholecystitis, and hepatic impairment was inconstant and moderate. All cases demonstrated a favorable outcome after the prompt initiation of intensive antibiotic therapy, with both renal abscesses and the acute acalculous cholecystitis receding without further complications.

A detailed description of the clinical condition is provided below.

Right upper quadrant pain was present in all four female patients with AAC. The pain increased upon pressure being applied with the ultrasound probe, producing a Murphy’s ultrasound sign. This sign has diagnostic importance in AAC.

Fever was present in all four patients with AAC (in three cases, it had values of 39 °C, and in one case, a value of 38 °C).

All four AAC patients presented lumbar pain: in three cases, the pain was on the side of the affected kidney, and in one case, renal pain was bilateral.

Leukocytosis was present in all four patients with ACC, with values between 13,000/mm^3^ and 19,000/mm^3^.

Leukocytosis was also present in all four cases of calculous cholecystitis (in the two cases of micro-calculous cholecystitis, the leukocyte count was 13,200/mm^3^ and 14,200/mm^3^, respectively).

High levels of aspartate aminotransferase (AST) were present in three out of four patients with AAC, and in two cases, they were accompanied by high levels of alanine aminotransferase (ALT). One patient presented increased vales of serum bilirubin concomitantly with high levels of AST and ALT.

In the two cases of micro-calculous cholecystitis, the liver enzymes as well as serum bilirubin were normal, being higher in only one of the two cases of calculous cholecystitis.

None of the four cases of ACC presented elevated serum creatinine despite the fact that one case suffered septic shock at presentation.

One of the two cases of micro-calculous cholecystitis and one case of calculous cholecystitis presented mild increases in their serum creatinine.

None of the cases of AAC had positive blood cultures.

None of the cases of AAC had positive urine cultures.

One of the two cases of micro-calculous cholecystitis presented a positive urine culture test, with over 100,000 *E. coli* CFU/mL, whilst the second one presented 50,000 *E. coli* CFU/mL. In one case, the blood cultures revealed coagulase-negative staphylococci. This case had a more severe evolution, with increases in the serum creatinine. 

One case with calculous cholecystitis presented E. Coli-positive blood cultures, with the respective urine culture test being positive for the same germ. This patient also presented increases in her serum creatinine with an acute kidney injury.

Notably, none of our four AAC cases presented complications; all cases had a favorable outcome.

None of the cases required surgical intervention.

All patients were administered an intravenous combination of two antibiotics.

In three cases, we used carbapenems (ertapenem in two cases and imipenem in one case) combined with an aminoglycoside, whilst in the fourth case, we used a third-generation cephalosporin, namely ceftriaxone, combined with an aminoglycoside. As far as aminoglycosides are concerned, we used amikacin in three cases and gentamicin in one case.

A contrast-enhanced CT scan of a patient showing kidney abscesses associated with ACC is presented in Figure 1.

CT is the best technique to evaluate the disease. MRI can be the modality of choice for patients allergic to iodine contrast media. Non-contrast axial CT image showing a right renal hypodense mass of fluid attenuation is indicative of disease. Contrast-enhanced axial CT image showing a right renal mass with more extensive areas of cortical hypoenhancement and cortical and subcapsular fluid collections is indicative of disease. Perinephric stranding and inflammation are present without right hydronephrosis.

## 4. Discussion

In our study of 67 patients with renal abscesses, associated acute cholecystitis was identified in eight (11.94%) of the cases.

Of these, four cases (5.97%) presented AAC, and four cases (5.97%) presented calculous cholecystitis, of which two presented biliary sludge (micro-calculous cholecystitis).

In patients with renal abscesses, Ganpathi et al. identified associated acute cholecystitis in 8.7% of cases [2].

The presence of AAC associated with a renal abscess raises the question of a possible dissemination of germs from the kidney to the gallbladder.

AAC is often encountered in intensive care units in critically ill patients. This is because it is a severe disease which is often accompanied by serious complications that require special therapeutic intervention. AAC can be related to surgery, trauma, burns, sepsis, shock, total parenteral nutrition, and prolonged fasting (Huffman and Schenker) [1].

AAC can occur in physically debilitated patients undergoing long-term hospitalization or in patients with complicated diabetes. Some of the risk factors for AAC are advanced age and cerebrovascular accidents (Gu et al.) [3]. Long-term stasis of the bile, with a thickening and change in its chemical composition, is considered to play an important role in the production of AAC (Ganpathi et al.) [2]. This situation is related to long-standing bed rest and malnutrition.

Microvascular lesions of the gallbladder are also described in relation to disorders of the circulatory system; Hakala et al. even mentioned their occurrence in microangiopathy [4]. An obstruction of the cystic duct is also considered to play an important role in the production of AAC (Howard) [5]. Sepsis, which accompanies many severe diseases, also plays a prominent role, as it allows the infection of the concentrated bile of the gallbladder to develop microvascular lesions.

Urinary tract infections (UTIs) are very common in medical practice. UTIs are frequently accompanied by bacteremia. According to Bahagon et al., 15% of patients with symptomatic UTIs are bacteremic at the time of presentation [6]. 

Renal abscesses represent a severe form of infection that allow for germ dissemination, either via the bloodstream or by spreading to the adjacent neighboring organs and tissues. Renal abscesses allow for the elimination of germs into the bloodstream for several days, with this elimination being sometimes intermittent.

Germs can colonize the gallbladder, even if favoring factors, such as biliary lithiasis, are absent, producing AAC. If an abscess is located in the right kidney, there is the additional possibility of direct spread to the adjacent neighboring organs and tissues.

Renal abscesses were found to be related to the liver’s involvement in producing hepatic abscesses (Vojinovic et al.) [7]. Tanwar et al. reported a case of pyonephrosis secondary to renal lithiasis which spread to the liver, causing a hepatic abscess [8]. A case of a right kidney abscess accompanied by the formation of a pyelo-hepatic fistula has also been described (Chung et al.) [9].

It is to be noted that currently, there are no descriptions of the spreading of renal abscesses to the gallbladder in the literature (Pub Med and Google Scholar).

As mentioned previously, AAC is frequently encountered in critically ill patients.

Jones et al. reported the association of AAC with infective endocarditis [10].

Renal abscesses also represent a severe form of infection that can be accompanied by the development of AAC.

Frequently, renal diseases spread to the retroperitoneum. Thus, traumatic lesions of the kidney are sometimes accompanied by a hemorrhage which extends retroperitoneally, producing a retroperitoneal hematoma. In the retroperitoneal space, this may represent the starting point of a urinoma. Renal abscesses can also spread retroperitoneally in the vicinity of the kidney, producing a perirenal abscess.

Although the attention around AAC has mainly focused on critically ill patients, more recently an increase in the number of AAC cases in the absence of the favoring factors found in severe diseases has been described. Thus, Parithivel et al. suggested that an important number of cases may appear de novo in patients without any predisposing factors [11]. By analyzing a 7-year case record of outpatients, Savoca et al. found that 77% developed the disease at home, without any evidence of an acute illness or trauma [12].

Our study identified the existence of AAC in a special category of patients with a severe infection, namely renal abscesses, adding renal abscesses to the etiology of AAC.

As far as the clinical picture of the patients is concerned, we found right upper quadrant pain in all four female patients with AAC. The pain increased upon the pressure applied with the ultrasound probe, producing a Murphy’s ultrasound sign. This sign has diagnostic importance in AAC. Ganpathi et al. mentioned the constant presence of this sign in AAC [2].

Fever was present in all four patients with AAC (in three cases, it had values of 39 °C, and in one case, of 38 °C). We must take into account that in our patients with AAC, infection at the level of the renal abscesses also accounted for the origin of a high fever.

All four AAC patients presented lumbar pain. In three cases, the pain was on the side of the affected kidney, and in one case, the renal pain was bilateral.

Leukocytosis was present in all four patients with ACC, with values between 13,000/mm^3^ and 19,000/mm^3^. Leukocytosis was also present in two cases of micro-calculous cholecystitis (13,200/mm^3^ and 14,200/mm^3^), respectively. Ganpathi et al. also found leukocytosis to be present in AAC [2].

High levels of aspartate aminotransferase (AST) were present in three out of four patients with AAC, and in two cases, they were accompanied by high levels of alanine aminotransferase (ALT). One patient presented increased values of serum bilirubin concomitantly with high levels of AST and ALT. Ganpathi et al. did not find increased levels of serum transaminases or serum bilirubin in any of the cases of AAC they analyzed [2].

In the two cases of micro-calculous cholecystitis, the liver enzymes as well as serum bilirubin were normal, only being higher in one of the two cases of calculous cholecystitis.

None of the four cases of ACC presented elevated serum creatinine, despite the fact that one case suffered septic shock at presentation. Taking into account that our patients had two foci of infection, one at the kidney level and one at the level of the gallbladder, one could have expected to find an acute kidney injury in our patients, though this proved not to be the case.

One of the two cases of micro-calculous cholecystitis and one of calculous cholecystitis presented mild increases in their serum creatinine.

It should be mentioned that AAC has also been reported in patients with a surgical acute kidney injury (Kes et al.) [13].

None of the cases of AAC had positive blood cultures.

None of the cases of AAC had positive urine cultures.

One of the two cases of micro-calculous cholecystitis presented a positive urine culture test, with over 100,000 Coliform CFU/mL, and the second one presented 50,000 germs/mL. In one case, the blood cultures revealed coagulase-negative staphylococci. This case had a more severe evolution, with increases in the serum creatinine.

One case with calculous cholecystitis presented E. Coli-positive blood cultures, with the respective urine culture test being positive for the same germ. This patient presented increases in her serum creatinine with an acute kidney injury.

The clinical condition of the patients required immediate antibiotic therapy with a combination of large-spectrum antibiotics; thus, the blood culture samples were limited to one test only, which was conducted at presentation. In fact, at least five blood culture tests are recommended in similar situations, but this requirement could not be met since an emergency antibiotic treatment had to be applied.

One must take into account the possibility that germ elimination at the level of the two foci of infection (renal and gallbladder) can be intermittent, partly explaining the negative blood and urine culture tests in our patients. Positive urine culture tests in renal abscesses can be related to the communication of an abscess with the urinary tract, which was not found in our patients.

Ultrasound plays a prominent role in diagnosing AAC. Computerized tomography is also useful for diagnosis. Cholescintigraphy proved to be the most sensitive diagnosis method. Mirvis et al. compared ultrasound, CT, and cholescintigraphy in diagnosing AAC, concluding that ultrasound had remarkable value [14]. Ultrasound has high sensitivity and specificity, being both inexpensive and easy to perform at the same time (Ganpathy et al.) [2]. In our study, ultrasound proved to be a powerful diagnostic tool. Computerized tomography was performed in only three patients, as cholescintigraphy was deemed unnecessary. Although ultrasound may sometimes present errors, it remains the main method for diagnosing AAC.

AAC can present complications during its evolution, such as gangrene, perforation, and abscesses (Wang et al.) [15].

Retroperitoneal abscesses and AAC have been reported after an iatrogenic colon injury (Dong et al.) [16].

The complications which occur in the evolution of AAC are higher in elderly patients (Wang et al.) [15].

Of note, none of our four AAC cases presented complications; they all had a favorable outcome.

None of the cases required surgical intervention.

All patients were administered an intravenous combination of two antibiotics.

In three cases, we used carbapenems (ertapenem in two cases and imipenem in one case) combined with aminoglycosides, and in one case, a third-generation cephalosporin, namely ceftriaxone, was combined with an aminoglycoside. As far as aminoglycosides are concerned, we used amikacin in three cases and gentamicin in one case.

It should be mentioned that antibiotic therapy commenced promptly and had favorable results in all four cases of AAC. The average duration of the therapy was 14 days.

An antibiotic treatment is usually administered to treat renal abscesses and only in the case of this being inefficient does one resort to surgical treatment (Rinder) [17].

For the treatment of acute cholecystitis, both acalculous and calculous, Capoor et al. recommend carbapenem or a combination of drugs for a severe infection [18].

In cases of severe AAC with complications, surgery is recommended, and cholecystectomy is sometimes performed

One has to mention the role of endoscopic ultrasound gallbladder drainage in severely ill patients with AAC (Mangiavillano et al.) [19]

Alghamdi HM. points to a case of non-surgical management in a case with acute acalculous cholecystitits perforation [20].

Our study points to the beneficial effects of medical therapy without surgical intervention in cases of AAC associated with renal abscesses.

## 5. Conclusions

AAC can be potentially associated with renal abscesses producing a severe clinical condition. All four of our AAC cases presented sepsis, and one case presented septic shock.

AAC can be associated with elevated liver enzymes and increased serum bilirubin. 

The renal function was normal in our patients with AAC.

The blood and urine cultures were negative in our patients with AAC.

The early use of antibiotic therapy did not allow for repeated blood culture tests and one cannot thus exclude the intermittent elimination of germs at the level of the infectious focus represented by the renal abscess.

All cases of AAC had a favorable course, with the resolution of both the renal abscess and of AAC after treatment using a combination of antibiotics.

None of the cases required surgery.

Our study emphasized the importance of the dual pathology of renal abscesses associated with AAC, which has so far not been assessed in the literature and needs to be explored further.

## Figures and Tables

**Figure 1 biomedicines-11-00632-f001:**
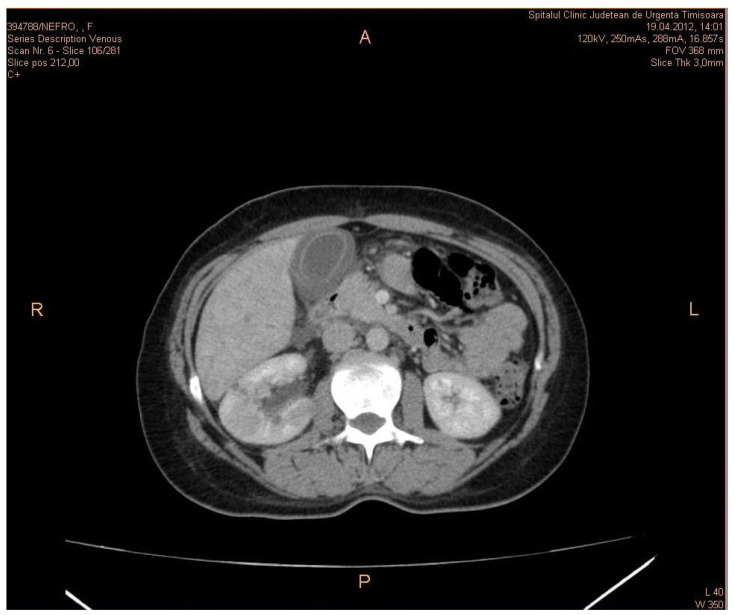
Contrast-enhanced CT scan of a patient in the venous phase showing kidney abscesses associated with acute acalculous cholecystitis.

## Data Availability

The data that support the findings of this study are available from the corresponding author upon reasonable request.

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
