# Peer review of "Acute Acalculous Cholecystitis Associated with Abscesses—An Unknown Dual Pathology"

_biomedicines, 2023, doi:10.3390/biomedicines11020632_

Round 1

Reviewer 1 Report

The manuscript presents the results of a retrospective analysis on 67 patients, with no statistical analysis. The conclusions could not be supported by the results. The article has serious flaws, in fact no methodology is mentioned. 

Author Response

Dear Reviewer,

Thank you very much for your insightful comments.

We have improved our paper accordingly.

  1. The paper has been sent out for English proofreading.
  2. Additionally, we used ChatGBT to modify the introduction and introduced two new references.
  3. It is true that our study is a department-wide retrospective cohort observational study. The nature of the data (imaging), as well as the lack of some data, such as C-reactive protein for all patients has seen us compelled to present that data in tabular format, we hereby attach. Also, this has not allowed us to compute the Tokyo score, as C-reactive protein was lacking in some patients (due to transient hospital based policy).
  4. Please see the paragraph below for supplemental data.
  5. Should there be any need for further revision of our paper, we would be glad to further improve our manuscript.
  6. Thank you for your insightful comments.

Reviewer 2 Report

- Please state the nature of the study (Prospective? Retrospective?)

- Please explain the majority of female patients in your study

- Results: delate "This section may be divided by subheadings. It should provide a concise and precise 104 description of the experimental results, their interpretation, as well as the experimental 105 conclusions that can be drawn."

- You have no control group therefore you cannot state in the conclusion that "ACC can be associated with renal abscesses producing a severe clinical condition". Please rephrase.

- The role of endoscopic ultrasound gallbladder drainage in severe ill patients with ACC should be mentioned. Doing so, cite PMID: 34339667 

Author Response

Dear Reviewer,

Thank you for your insightful comments.

We have amended our paper accordingly.

  1. Our study is a department-wide retrospective cohort observational study.
  2. As an observational study, the majority of patients of female gender was an incidental finding; we considered it however concordant with data  found in the literature, namely the canonical F`s which still hold true: female, forty, fat, fertile, family history.
  3. We have deleted the suggested sentence.
  4. We have rephrased the suggested item.
  5. We have added the requested mention of the role of endoscopic ultrasound gallbladder drainage in severe ill patients with ACC, by citing PMID: 34339667 

Reviewer 3 Report

Please indicate how to diagnose a renal abscess. You should show culture results (urine and bile) for cholecystitis complicated by renal abscess. Please describe the severity of cholecystitis (Tokyo criteria). It is unclear whether renal abscess was complicated by cholecystitis or cholecystitis by renal abscess. Please consider the mechanism of these occurrences.

Author Response

Dear Reviewer,

Thank you for your insightful comments.

1. I have asked our imaging specialist and here is what she replied:

CT is the best technique to evaluate the disease. MRI can be the modality of choice for patients allergic to iodine contrast media. Non-contrast axial CT image showing a right renal hypodense mass of fluid attenuation is indicative of disease. Contrast-enhanced axial CT image showing right renal mass with more extensive areas of cortical hypoenhancement and cortical and subcapsular fluid collections is indicative of disease. Perinephric stranding and inflammation is present   without right hydronephrosis. 2. The lack of C-reactive protein in all patients due to hospital based reimbursement policy precluded us from computing the Tokyo score. Some of the requested data are shown in tabular format in the attachment, kindly see attachment). 3. We posit that our findings represent rare hepato-biliary complications of renal abscesses in nephrology; this is just our point of view, but of course the reverse might also hold true. 

Round 2

Reviewer 1 Report

According to all the changes, the quality of the manuscript significantly improved. Thus, I recommend publication in this form.

Reviewer 2 Report

I have no further comments 

Reviewer 3 Report

No additional comments.